

# Dissection of three quantitative trait loci for grain size on the long arm of chromosome 10 in rice (*Oryza sativa* L.)

Yu-Jun Zhu[1,2], Zhi-Chao Sun[1], Xiao-Jun Niu[1], Jie-Zheng Ying[1], Ye-Yang Fan[1], Tong-Min Mou[2], Shao-Qing Tang[1] and Jie-Yun Zhuang[1]

[1] State Key Laboratory of Rice Biology and Chinese National Center for Rice Improvement, China National Rice Research Institute, Hangzhou, China
[2] State Key Laboratory of Crop Genetic Improvement and National Center of Plant Gene Research, Huazhong Agricultural University, Wuhan, China

Corresponding authors
Shao-Qing Tang,
tangshaoqing@caas.cn
Jie-Yun Zhuang,
zhuangjieyun@caas.cn

## ABSTRACT

**Background:** Thousand grain weight is a key component of grain yield in rice, and a trait closely related to grain length (GL) and grain width (GW) that are important traits for grain quality. Causal genes for 16 quantitative trait loci (QTL) affecting these traits have been cloned, but more QTL remain to be characterized for establishing a genetic regulating network. A QTL controlling grain size in rice, *qGS10*, was previously mapped in the interval RM6100–RM228 on chromosome 10. This study aimed to delimitate this QTL to a more precise location.

**Method:** A total of 12 populations were used. The ZC9 population comprised 203 $S_{1:2}$ families derived from a residual heterozygous (RH) plant in the $F_9$ generation of the *indica* rice cross Teqing (TQ)/IRBB52, segregating the upper region of RM6100–RM228 and three more regions on chromosomes 1, 9, and 11. The Ti52-1 population comprised 171 $S_1$ plants derived from one RH plant in $F_7$ of TQ/IRBB52, segregating a single interval that was in the lower portion of RM6100–RM228. The other ten populations were all derived from Ti52-1, including five $S_1$ populations with sequential segregating regions covering the target region and five near isogenic line (NIL) populations maintaining the same segregating pattern. QTL analysis for 1,000-grain weight, GL, and GW was performed using QTL IciMapping and SAS procedure GLM.

**Result:** Three QTL were separated in the original *qGS10* region. The *qGL10.1* was located in the upper region RM6704–RM3773, shown to affect GL only. The *qGS10.1* was located within a 207.1-kb interval flanked by InDel markers Te20811 and Te21018, having a stable and relatively high effect on all the three traits analyzed. The *qGS10.2* was located within a 1.2-Mb interval flanked by simple sequence repeat markers RM3123 and RM6673. This QTL also affected all the three traits but the effect was inconsistent across different experiments. QTL for grain size were also detected in all the other three segregating regions.

**Conclusion:** Three QTL for grain size that were tightly linked on the long arm of chromosome 10 of rice were separated using NIL populations with sequential segregating regions. One of them, *qGS10.1*, had a stable and relatively high effect on grain weight, GL, and GW, providing a good candidate for gene cloning. Another QTL, *qGS10.2*, had a significant effect on all the three traits but the effect was inconsistent across different experiments, providing an example of genotype-by-environmental interaction.

## INTRODUCTION

Thousand grain weight (TGW) is a key component of grain yield in rice, and a trait closely related to grain length (GL) and grain width (GW) that are important traits for grain quality. These traits are all complex and inherited quantitatively. To date, causal genes for 16 quantitative trait loci (QTL) affecting these traits have been isolated. They are distributed across eight of the 12 chromosomes of rice. Two of them, *GW7/GL7* and *GS9* (*Wang et al., 2015a*, *2015b*; *Zhao et al., 2018*), have large effect on grain shape but no effect on TGW due to opposite allelic directions of the effects on GL and GW. Four of them, *GW2* (*Song et al., 2007*), *GS5* (*Li et al., 2011*), *GSE5* (*Duan et al., 2017*), and *GW8* (*Wang et al., 2012*), have larger effects on TGW and GW than on GL. Six others, *GL2/GS2* (*Che et al., 2016*; *Hu et al., 2015*), *GS3* (*Fan et al., 2006*), *GL3.1/qGL3* (*Qi et al., 2012*; *Zhang et al., 2012*), *OsLG3* (*Yu et al., 2017*), *qTGW3* (*Hu et al., 2018*), and *GW6a* (*Song et al., 2015*), have larger effects on TGW and GL than on GW. The remaining four, *OsLG3b* (*Yu et al., 2018*), *GL4* (*Wu et al., 2017*), *TGW6* (*Ishimaru et al., 2013*), and *GLW7* (*Si et al., 2016*), have large effects on TGW and GL but no effect on GW. These genes encode proteins that are involved in regulating cell proliferation and elongation, including phytohormones, transcriptional regulatory factors and components of pathways such as ubiquitin-proteasome, G protein signaling and MAPK signaling (*Li et al., 2018*). These studies have substantially improved our understanding of the molecular basis of grain weight in rice, but the information is rather fragmental and more genes remain to be isolated and characterized (*Zuo & Li, 2014*).

In genetic analysis of agronomic traits, close linkage of QTL controlling the same trait has been frequently observed. Cloned QTL having major effects on grain weight and grain size in rice have provided clear examples. *OsLG3* and *OsLG3b* for GL (*Yu et al., 2017*, *2018*) were located in a 1.7-Mb region of chromosome 3, *GS5* and *GSE5* for GW in a 1.9-Mb region of chromosome 5 (*Li et al., 2011*; *Duan et al., 2017*), and *TGW6* and *GW6a* for grain weight in a 0.6-Mb region of chromosome 6 (*Ishimaru et al., 2013*; *Song et al., 2015*). Similar results have also been reported for QTL having minor effects on these traits. Targeting at a QTL region affecting grain weight difference between two *indica* rice varieties, six minor-effect QTL were separated in a 7.1-Mb interval on chromosome 1, including *qTGW1.1a*, *qTGW1.1b*, *qTGW1.2a*, *qTGW1.2b*, *qGS1-35.2*, and *qGW1-35.5* (*Wang et al., 2015c*; *Zhang et al., 2016*; *Dong et al., 2018*). Assumably, validation and dissection of QTL that were previously coarsely mapped could provide a large number of candidates for isolating individual genes controlling grain weight and grain size in rice.

Rice is a staple food for about half of the world's population. Large scale cultivation of hybrid rice has made a great contribution to ensuring the food security in China and may make contributions in many other countries as well (*Yuan, 2014*; *Nalley et al., 2016*). Minor-effect QTL is believed to play a critical role in controlling genetic variations of yield traits among commercial rice varieties and the dissection of these QTL would

**Table 1 Rice populations and field experiments.**

| Population | | | Segregating regions on chromosome 10 | Sample | Location and growth period |
|---|---|---|---|---|---|
| Name | Type | Generation | | | |
| For validation of qGS10 | | | | | |
| ZC9 | $S_{1:2}$ | $F_{10:11}$ | RM6704–RM3773 | 203 lines | HZ: May–September 2017 |
| Ti52-1 | $S_1$ | $F_8$ | RM25766–RM228 | 171 plants | LS: December 2015–April 2016 |
| For dissection of qGS10 | | | | | |
| G10-1 | $S_1$ | $F_{10}$ | RM25766–Te20811 | 278 plants | LS: December 2016–April 2017 |
| G10-2 | $S_1$ | $F_{10}$ | Te20863–Te20993 | 172 plants | LS: December 2016–April 2017 |
| G10-3 | $S_1$ | $F_{10}$ | Te20863–RM228 | 289 plants | LS: December 2016–April 2017 |
| G10-4 | $S_1$ | $F_{10}$ | Te21185–RM228 | 292 plants | LS: December 2016–April 2017 |
| G10-5 | $S_1$ | $F_{10}$ | RM25845–RM228 | 343 plants | LS: December 2016–April 2017 |
| G11-1 | NIL | $F_{10:11}$ | RM25766–Te20811 | 29 : 29 lines | HZ: May–September 2017 |
| G11-2 | NIL | $F_{10:11}$ | Te20863–Te20993 | 29 : 28 lines | HZ: May–September 2017 |
| G11-3 | NIL | $F_{10:11}$ | Te20863–RM228 | 30 : 35 lines | HZ: May–September 2017 |
| G11-4 | NIL | $F_{10:11}$ | Te21185–RM228 | 30 : 30 lines | HZ: May–September 2017 |
| G11-5 | NIL | $F_{10:11}$ | RM25845–RM228 | 38 : 38 lines | HZ: May–September 2017 |

**Note:**

NIL, near isogenic line; HZ, Hangzhou, Zhejiang Province; LS, Lingshui, Hainan Province. For the five NIL populations, numbers of NILs carrying the Teqing and IRBB52 homozygous segments are indicated before and after the colon, respectively.

facilitate the development of high-yielding rice varieties (*Kinoshita et al., 2017*). In our previous studies, QTL analysis for grain size was performed using three recombinant inbred line populations constructed from crosses between parental lines of commercial hybrid rice. One QTL, *qGS10* conferring grain size difference between Teqing (TQ) and IRBB lines was mapped in the interval RM6100–RM228 on chromosome 10 and then validated using near isogenic lines (NILs) of the same cross (*Wang et al., 2017*). In the present study, NIL populations segregated in sequential order around the *qGS10* region were constructed from the TQ/IRBB52 cross and used for QTL analysis. Three tightly-linked QTL were separated. Among them, *qGS10.1* had a stable and relatively high effect on grain weight, GL, and GW, *qGS10.2* had significant effects on all the three traits but the effects appeared to be greatly influenced by genotype-by-environmental interaction, and *qGL10.1* was shown to affect GL only.

# MATERIALS AND METHODS

## Rice materials

A total of 12 populations of rice (*Oryza sativa* L.) were used (Table 1). As illustrated in Fig. 1 and described below, these populations were derived from two residual heterozygous plants of the *indica* rice cross TQ/IRBB52, namely ZC9 and Ti52-1. ZC9 was an $F_9$ plant carrying four heterozygous segments, including RM6704–RM3773 on chromosome 10 and three others on chromosomes 1, 9, and 11 (Fig. 2). Ti52-1 was an $F_7$ plant carrying one heterozygous segment, that is, RM25766–RM228 on chromosome 10. The intervals RM6704–RM3773 and RM25766–RM228 are adjacent and jointly covered the

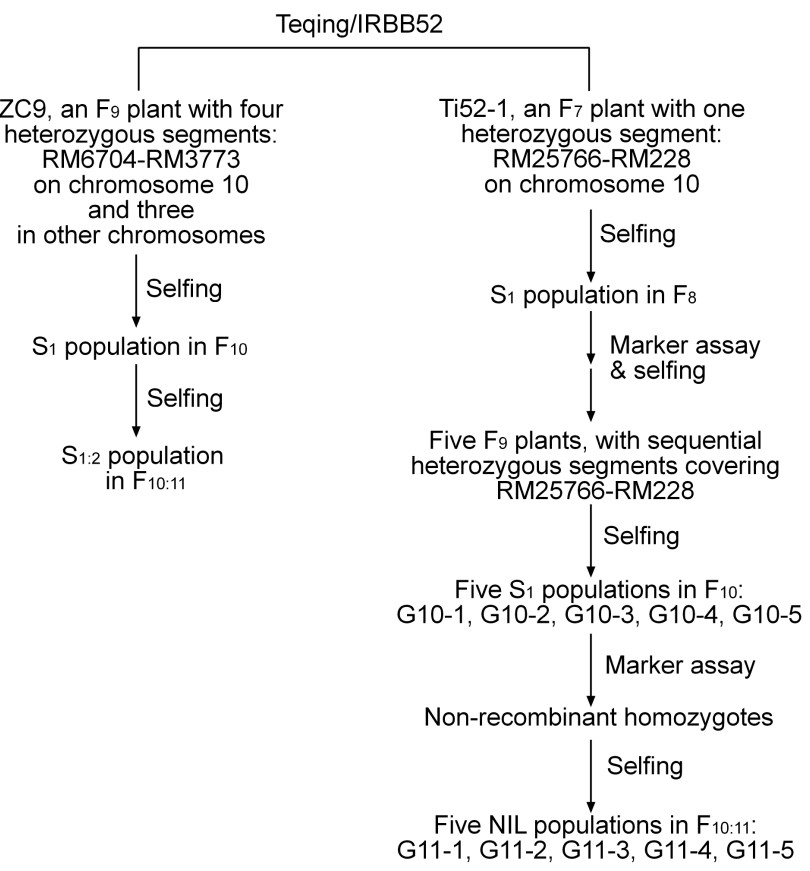

**Figure 1 Development of the rice populations used in this study.** NIL, near isogenic line.

RM6100–RM228 region previously mapped for *qGS10* (*Wang et al., 2017*). Two populations were constructed and used for QTL mapping. One consisted of 203 $S_{1:2}$ families derived from ZC9, and the other comprised 171 $S_1$ plants produced from Ti52-1 (Table 1).

The other 10 populations used for QTL mapping were all derived from Ti52-1. Five $F_9$ plants with sequential heterozygous regions covering RM25766–RM228 were selected from progenies of Ti52-1. They were selfed to produce five new $S_1$ populations. The number of plants in each population ranged from 172 to 343 (Table 1). In each population, non-recombinant homozygotes were selected and selfed. Five NIL populations in $F_{10:11}$ were established, in which the numbers of NILs having the same genotype ranged from 28 to 38 (Table 1).

## Field trial and phenotypic evaluation

The filed experiments were conducted at the experimental stations of the China National Rice Research Institute located in either Hangzhou, Zhejiang Province, or Lingshui, Hainan Province, China. The plants were grown with 16.7 cm between plants and 26.7 cm between rows.

The six $S_1$ populations were grown without replication and the plants were harvested individually. The remaining six populations, including one $S_{1:2}$ population and five NIL

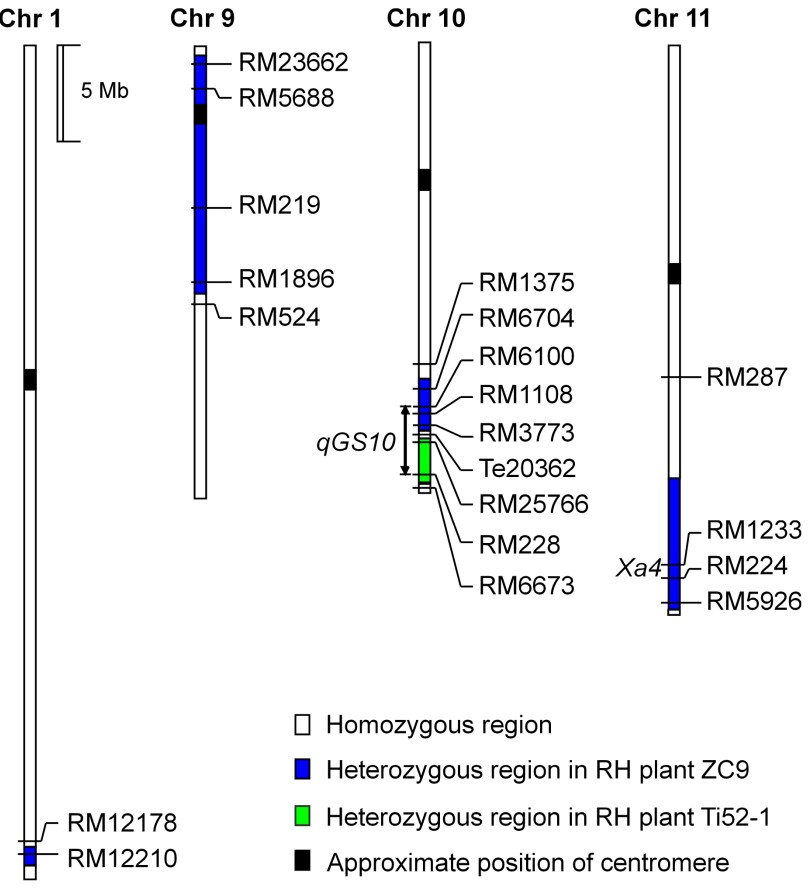

**Figure 2 Heterozygous regions of the residual heterozygous plants ZC9 and Ti52-1.**

populations, were tested using a randomized complete block design with two replications. A total of 12 plants per line were planted in each row. The middle five plants were harvested. Measurement of TGW, GL, and GW in all the experiments followed the method of *Zhang et al. (2016)*.

## DNA marker analysis

For the ZC9 population, marker data and genetic maps have been available (*Sun et al., 2018*). For the other populations, a total of 12 DNA markers were used, including five simple sequence repeat (SSR) and seven InDel markers. The SSRs were selected from the Gramene database (www.gramene.org), while the InDels (Table 2) were developed based on the sequence differences between TQ and IRBB52. DNA extraction and PCR amplification followed the method of *Zheng et al. (1995)* and *Chen et al. (1997)*, respectively.

## Data analysis

Mapmarker/Exp 3.0 was performed to calculate the genetic distance which was presented in centiMorgan using the Kosambi function (*Lander et al., 1987*). For the six $S_1$ populations and the ZC9 $S_{1:2}$ population, QTL analysis was performed with the inclusive

**Table 2 Polymorphic markers developed and used in this study.**

| Name | Forward primer (5′-3′) | Reverse primer (5′-3′) |
| --- | --- | --- |
| Te20362 | TCTACCCTCACCTCGCGTA | TCCATTTCTAATCCACGACCT |
| Te20811 | ATGGAGAAGTAGCAGAGTACATT | TGGATCATCAAAGGCTCACAAA |
| Te20863 | GCCGCCTCTACGAGTT | AAAATCACCCGATTACCACAG |
| Te20993 | AAAATTTAACTTCTGCATGTTG | TCGAGCTTGCATGTCATC |
| Te21018 | GCTTCTAAACTGCTAACAGGT | GTCATTACAATTGCACATAGGAG |
| Te21052 | TCTGAATATTAGCATAGCCGAGT | CTTTTCCTGGGTATATGGAAC |
| Te21185 | ACCGCCGATGACATGC | TCTTCTTCGATCGGGTC |

composite interval mapping (ICIM) using the BIP (QTL mapping in the biparental populations) functionality of QTL IciMapping 4.1 (*Meng et al., 2015*). *LOD* thresholds were calculated with 1,000 permutation tests ($P < 0.05$) and used to declare a putative QTL. For the five NIL populations, SAS procedure GLM (*SAS Institute Inc, 1999*) was used to determine phenotypic differences in each population. A mixed model GENOTYPE + LINE (GENOTYPE) + REP + GENOTYPE*REP was applied, in which LINE (GENOTYPE) was defined as a random effect and used as the error term to test GENOTYPE differences (*Dai et al., 2008*). Additive effect and the proportion of phenotypic variance explained ($R^2$) were estimated with the same model. QTL were designated following the rule proposed by *McCouch & CGSNL (2008)*.

## RESULTS

### QTL for grain size detected in the ZC9 and Ti52-1 populations

The ZC9 $S_{1:2}$ population was segregated in the interval RM6704–RM3773 that covered the upper portion of the *qGS10* region, and in three other intervals (Fig. 2). All the four regions were found to have significant effects on one or two of the three grain-size traits analyzed (Table 3). In the interval RM6704–RM3773 on chromosome 10, the effects were significant on GL but non-significant on the other two traits. This QTL explained 9.00% of the phenotypic variance and the IRBB52 allele increased GL by 0.020 mm. The *qGL1* and *qGW1* associated with RM12210 on chromosome 1 had $R^2$ values of 32.71% and 5.84%, respectively, with the IRBB52 allele increased GL by 0.041 mm and decreased GW by 0.007 mm. The *qTGW9* and *qGW9* in the interval RM219–RM1896 on chromosome 9 had $R^2$ values of 19.15% and 7.42%, respectively, with the IRBB52 allele increased TGW by 0.16 g and GW by 0.007 mm. The *qTGW11* and *qGW11* in the interval RM224–RM5926 on chromosome 11 had $R^2$ values of 13.81 and 31.69%, respectively, with the IRBB52 allele decreased TGW by 0.14 g and GW by 0.016 mm.

The Ti52-1 population was segregating in the interval RM25766–RM228 only, which covered the lower portion of the *qGS10* region (Fig. 2). This region was found to have significant effects on all the three grain-size traits analyzed (Table 3). The QTL effects contributed 20.15–36.58% to the phenotypic variance of each trait, with the TQ allele increasing TGW, GL, and GW by 0.59 g, 0.081 mm, and 0.024 mm, respectively. Since *qGS10* was previously reported to affect all these three traits with

**Table 3  QTL for TGW, GL, and GW detected in the ZC9 and Ti52-1 populations.**

| Population | Segregating region | | Trait | QTL | LOD | A | D | $R^2$ (%) |
|---|---|---|---|---|---|---|---|---|
| | Chr | Interval | | | | | | |
| ZC9 | 1 | RM12210 | GL | qGL1 | 19.1 | 0.041 | −0.007 | 32.71 |
| | | | GW | qGW1 | 4.5 | −0.007 | −0.001 | 5.84 |
| | 9 | RM219-RM1896 | TGW | qTGW9 | 10.5 | 0.16 | −0.05 | 19.15 |
| | | | GW | qGW9 | 5.6 | 0.007 | −0.004 | 7.42 |
| | 10 | RM6100-RM1108 | GL | qGL10.1 | 6.1 | 0.020 | 0.003 | 9.00 |
| | 11 | RM224-RM5926 | TGW | qTGW11 | 7.8 | −0.14 | 0.023 | 13.81 |
| | | | GW | qGW11 | 20.2 | −0.016 | 0.003 | 31.69 |
| Ti52-1 | 10 | RM25766–RM228 | TGW | qTGW10 | 10.1 | −0.59 | −0.11 | 24.84 |
| | | | GL | qGL10.2 | 16.7 | −0.081 | 0.011 | 36.58 |
| | | | GW | qGW10.2 | 8.3 | −0.024 | −0.003 | 20.15 |

**Note:**
TGW, 1,000-grain weight (g); GL, grain length (mm); GW, grain width (mm); Chr, chromosome; A, additive effect of replacing a Teqing allele with an IRBB52 allele; D, dominance effect; $R^2$, proportion of phenotypic variance explained by the QTL effect. QTL were designated following the rule proposed by *McCouch & CGSNL (2008)*.

the TQ allele always increasing trait values (*Wang et al., 2017*), it is concluded that *qGS10* was located in the segregating region of the Ti52-1 population, and another QTL, *qGL10.1*, was located in an adjacent interval that was segregated in the ZC9 population (Fig. 3A).

## Dissection of *qGS10* into two QTL for grain size and grain weight

For further validation and dissection of *qGS10*, five $S_1$ populations with sequential segregating regions covering the interval RM25766–RM228 were established (Table 1; Fig. 3B). Results of QTL analysis for TGW, GL, and GW using these populations are shown in Table 4. Significant QTL effects were detected in G10-2, G10-4, and G10-5 but not in the other two populations. In G10-2, the effects were significant for all the three traits. The TQ allele increased TGW, GL, and GW by 0.19 g, 0.030 mm, and 0.004 mm, respectively. In G10-4 and G10-5, significant QTL effects were detected for GL only. The enhancing alleles were derived from IRBB52 in both populations, with similar additive effects of 0.019 and 0.023 mm, and similar $R^2$ values of 5.11% and 3.01%. As shown in Fig. 3B, the region segregated in G10-2 was separated from those segregated in G10-4 and G10-5, indicating that at least two QTL were responsible for the grain-size variations in these populations. One QTL was located in the segregating region of G10-2 and another QTL may be located in the common segregating region of G10-4 and G10-5. To validate these results, non-recombinant homozygotes of the five $S_1$ populations were identified, from which five NIL populations maintaining the original sequential segregating regions were constructed (Table 1; Fig. 3B).

Distributions of TGW, GL, and GW in the five NIL populations are shown in Fig. 4. The three traits were continuously distributed in all the populations, but differentiation between the two genotypic groups was observed in four populations. Concentration of the TQ and IRBB52 homozygous lines, respectively, in the high- and low-value regions was

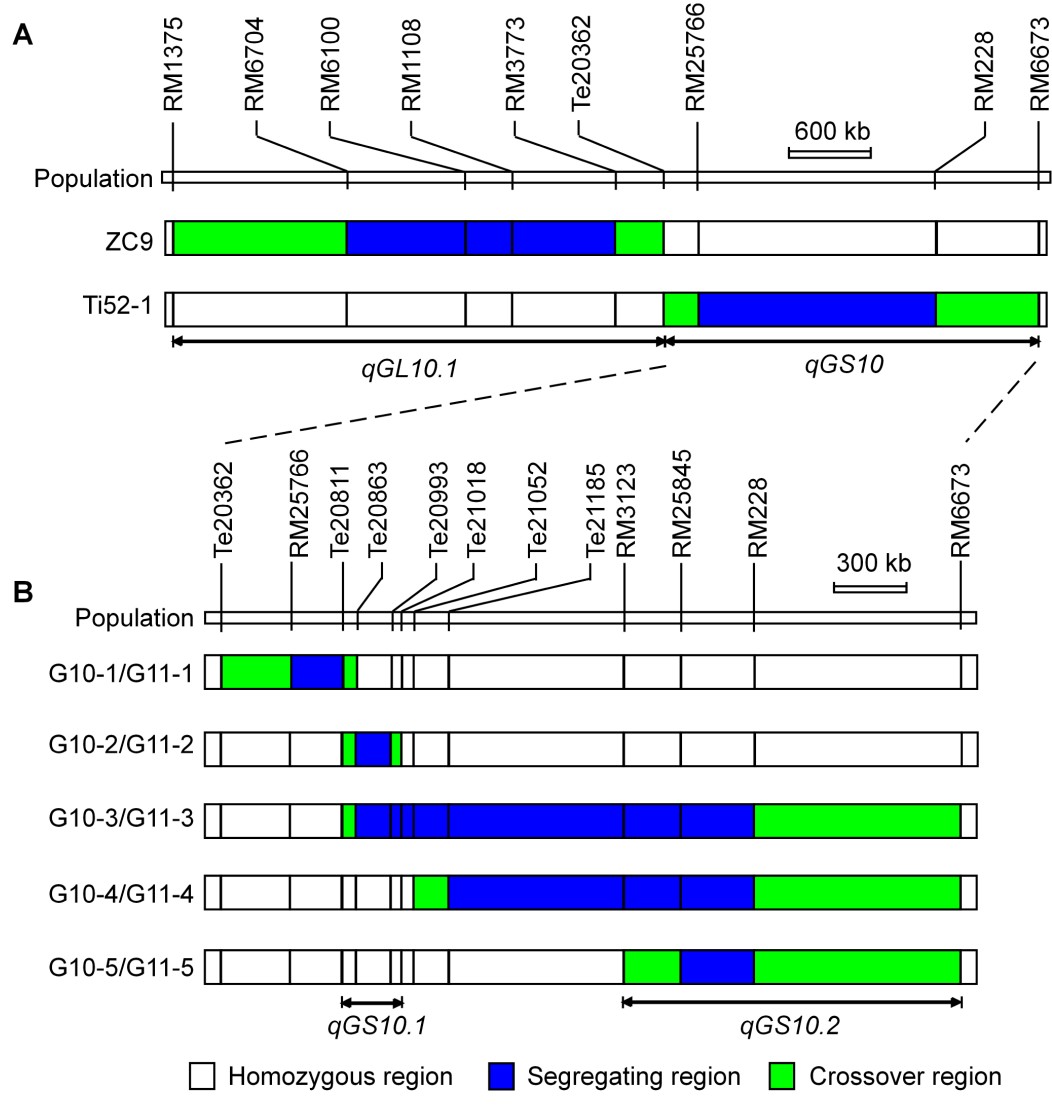

**Figure 3 Genotypic composition of the rice populations in the target region on chromosome 10.**
(A) ZC9, a $S_{1:2}$ population in $F_{10:11}$. Ti52-1, a $S_1$ population in $F_8$. (B) G10-1, G10-2, G10-3, G10-4 and G10-5, five $S_1$ populations in $F_{10}$. G11-1, G11-2, G11-3, G11-4 and G11-5, five near isogenic lines populations in $F_{10:11}$.

evident for TGW and GL in G11-2, and for all the three traits in G11-3. The same tendency, though less obvious, was found for TGW and GW in G11-4 and G11-5. Since the segregating region of G11-3 covered the segregating regions of G11-2, G11-4, and G11-5 (Fig. 3B), these results suggest that allelic differences of the two QTL, one located in the segregating region of G11-2 and the other in the common segregating region of G11-4 and G11-5, could be the main cause of phenotypic variations in the five NIL populations.

Results of two-way ANOVA for TGW, GL, and GW using the five NIL populations are presented in Table 5. In G11-1, no significant phenotypic difference was detected between the two genotypic groups, matching well to the detection of no QTL in its origin population G10-1. In G11-2, the two genotypic groups showed significant differences on

**Table 4 QTL for TGW, GL, and GW detected in five $S_1$ populations.**

| Population | Segregating region | Trait | LOD | A | D | $R^2$ (%) |
|---|---|---|---|---|---|---|
| G10-1 | RM25766–Te20811 | TGW | n.s. | | | |
| | | GL | n.s. | | | |
| | | GW | n.s. | | | |
| G10-2 | Te20863–Te20993 | TGW | 14.6 | −0.19 | 0.66 | 32.09 |
| | | GL | 28.6 | −0.030 | 0.140 | 53.10 |
| | | GW | 8.5 | −0.004 | 0.035 | 20.12 |
| G10-3 | Te20863–RM228 | TGW | n.s. | | | |
| | | GL | n.s. | | | |
| | | GW | n.s. | | | |
| G10-4 | Te21185–RM228 | TGW | n.s. | | | |
| | | GL | 2.7 | 0.019 | 0.023 | 5.11 |
| | | GW | n.s. | | | |
| G10-5 | RM25845–RM228 | TGW | n.s. | | | |
| | | GL | 2.2 | 0.023 | −0.003 | 3.01 |
| | | GW | n.s. | | | |

Note:
TGW, 1,000-grain weight (g); GL, grain length (mm); GW, grain width (mm); A, additive effect of replacing a Teqing allele with an IRBB52 allele; D, dominance effect; $R^2$, proportion of phenotypic variance explained by the QTL effect; n.s., not significant.

each trait. The TQ allele increased TGW, GL, and GW by 0.21 g, 0.032 mm, and 0.004 mm, with $R^2$ values of 40.58%, 28.55%, and 3.99%, respectively. These results were in general agreement with QTL detected in the G10-2 population from which G11-2 was originated, confirming that a QTL affecting TGW, GL, and GW was located in the segregating region of G10-2 and G11-2. This QTL, designated *qGS10.1*, was mapped within the region flanked by InDel markers Te20811 and Te21018 (Fig. 3B), corresponding to a 207.1-kb region in the Nipponbare genome.

In G11-4 and G11-5 which were segregated in homozygous regions of G11-2, significant QTL effects were detected for TGW and GW with the TQ allele always increasing the trait values. The additive effects were 0.13 and 0.12 g for TGW, and 0.012 and 0.011 mm for GW. These results indicate that a QTL for TGW and GW was located in the common segregating region of G11-4 and G11-5. This QTL, designated *qGS10.2*, was mapped within the region flanked by SSR markers RM3123 and RM6673 (Fig. 3B), corresponding to a 1.2-Mb region in the Nipponbare genome. It is noted that the effect of *qGS10.2* on TGW and GW was not detected in the G10-4 and G10-5 populations from which G11-4 and G11-5 were originated. For the other trait GL, a small effect with the enhancing allele derived from IRBB52 was detected in G11-5, which was in accordance with the effects detected in G10-4 and G10-5.

For the remaining NIL population, G11-3, significant QTL effects with the enhancing allele derived from TQ were detected for all the three traits. The additive effects and $R^2$ values estimated for TGW and GW were higher in G11-3 than in G11-2, G11-4, and G11-5. This is reasonable because *qGS10.1* and *qGS10.2* have the same allelic direction and the segregating region of G11-3 covered both QTL.

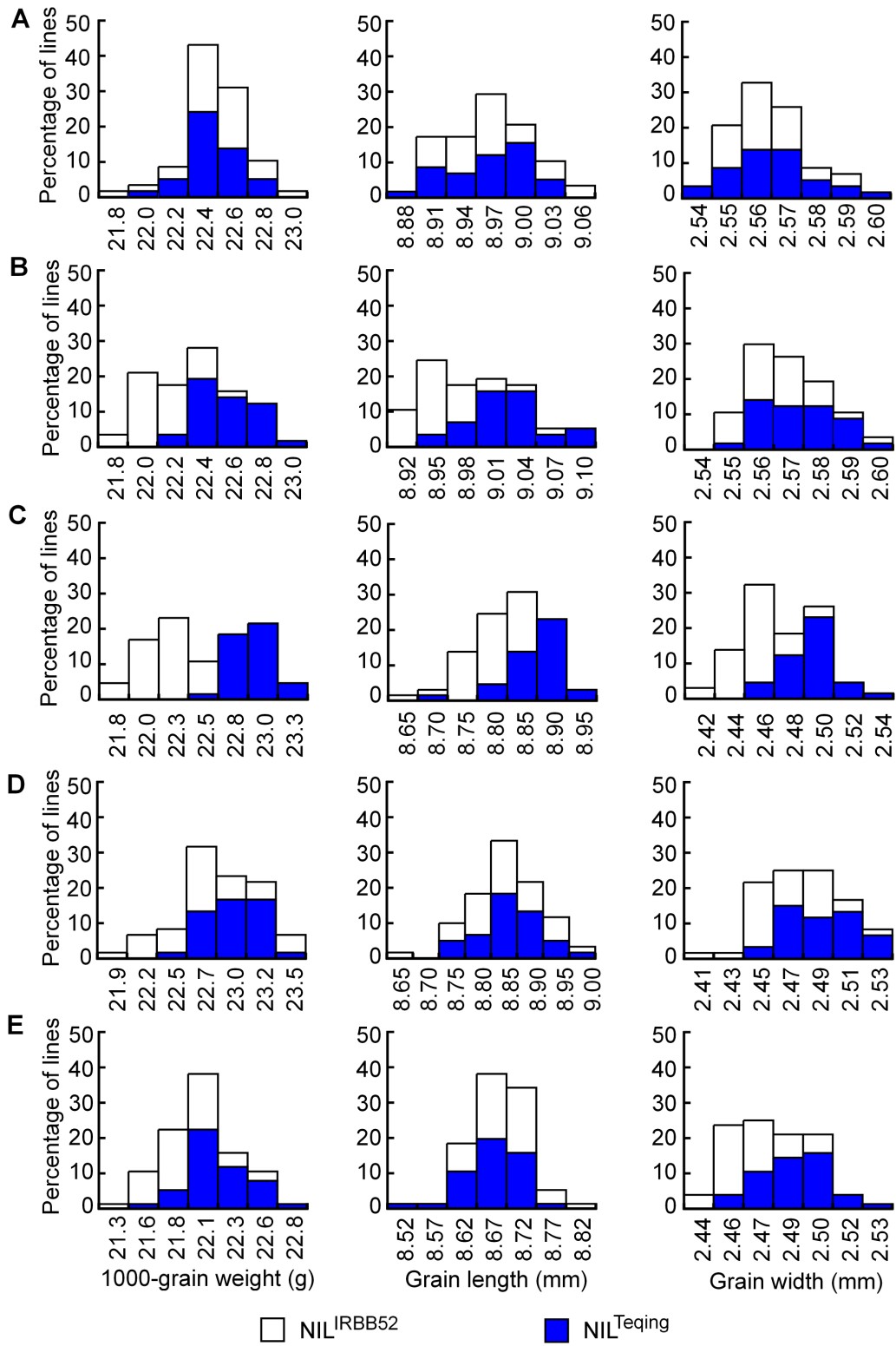

**Figure 4 Distributions of 1,000-grain weight, grain length, and grain width in the five NIL populations.** (A) G11-1. (B) G11-2. (C) G11-3. (D) G11-4. (E) G11-5.

**Table 5 QTL for TGW, GL, and GW detected in five NIL populations.**

| Name | Segregating region | Trait | Phenotypic mean | | P | A | $R^2$ (%) | QTL segregated |
|------|--------------------|-------|-----------------|--|---|---|-----------|----------------|
| | | | $NIL^{TQ}$ | $NIL^{IRBB52}$ | | | | |
| G11-1 | RM25766–Te20811 | TGW | 22.37 ± 0.18 | 22.36 ± 0.25 | 0.9593 | | | No QTL |
| | | GL | 8.953 ± 0.041 | 8.956 ± 0.044 | 0.7746 | | | |
| | | GW | 2.560 ± 0.015 | 2.559 ± 0.011 | 0.6110 | | | |
| G11-2 | Te20863–Te20993 | TGW | 22.46 ± 0.21 | 22.03 ± 0.19 | <0.0001 | −0.21 | 40.58 | qGS10.1 |
| | | GL | 9.009 ± 0.041 | 8.946 ± 0.038 | <0.0001 | −0.032 | 28.55 | |
| | | GW | 2.568 ± 0.013 | 2.561 ± 0.012 | 0.0309 | −0.004 | 3.99 | |
| G11-3 | Te20863–RM228 | TGW | 22.80 ± 0.16 | 22.06 ± 0.23 | <0.0001 | −0.37 | 63.80 | qGS10.1+qGS10.2 |
| | | GL | 8.845 ± 0.046 | 8.767 ± 0.045 | <0.0001 | −0.039 | 27.34 | |
| | | GW | 2.485 ± 0.019 | 2.446 ± 0.018 | <0.0001 | −0.020 | 40.55 | |
| G11-4 | Te21185–RM228 | TGW | 22.89 ± 0.25 | 22.63 ± 0.41 | <0.0001 | −0.13 | 17.59 | qGS10.2 |
| | | GL | 8.839 ± 0.058 | 8.823 ± 0.075 | 0.4628 | | | |
| | | GW | 2.483 ± 0.021 | 2.460 ± 0.024 | <0.0001 | −0.012 | 24.56 | |
| G11-5 | RM25845–RM228 | TGW | 22.04 ± 0.27 | 21.80 ± 0.28 | <0.0001 | −0.12 | 10.32 | qGS10.2 |
| | | GL | 8.645 ± 0.054 | 8.664 ± 0.046 | 0.0356 | 0.010 | 2.29 | |
| | | GW | 2.480 ± 0.017 | 2.459 ± 0.017 | <0.0001 | −0.011 | 17.51 | |

**Note:**

TGW, 1,000-grain weight (g); GL, grain length (mm); GW, grain width (mm); A, additive effect of replacing a Teqing allele with an IRBB52 allele; $R^2$, proportion of phenotypic variance explained by the QTL effect. $NIL^{TQ}$ and $NIL^{IRBB52}$ are near-isogenic lines with Teqing and IRBB52 homozygous genotypes in the segregating region, respectively.

# DISCUSSION

Grain yield of $F_1$ hybrid rice are known to be positively correlated to both the $F_1$ heterosis and the parental values. Regarding phenotypic performances of the three key yield components, the highest correlation between restorer line (male parent) and $F_1$ hybrid was found for grain weight (*Xiang et al., 2016*). Identification of QTL responsible for grain weight variation among important restorer lines would provide new information on understanding the genetic basis of grain yield in rice. In this study, QTL analysis using NILs derived from a cross between rice restorer lines TQ and IRBB52 resulted in the dissection of three QTL for grain weight and grain size that were tightly lined on the long arm of chromosome 10, providing new candidates for gene cloning and marker assisted selection.

No QTL for grain weight and grain size detected on chromosome 10 have been cloned, but a number of studies (*Ren et al., 2016*; *Qi et al., 2017*; *Okada et al., 2018*) have located QTL for these traits in the *qGS10* region. In addition, this region was associated with grain-size difference not only between restorer lines TQ and IRBB52, but also between Zhenshan 97 and Milyang 46 (*Wang et al., 2017*) that were among the most important female and male parents of commercial hybrid rice, respectively. One of the three QTL for grain size separated in this region, *qGS10.1*, was shown to have a stable and relatively high effect on all the three grain-size traits analyzed. Cloning of this QTL may identify an important gene controlling grain yield of modern rice varieties.

Genotype-by-environmental interaction is an important factor influencing the performance of complex traits. The effect of another QTL affecting all the three traits in this study, *qGS10.2*, appeared to be inconsistent across different experiments. In the $S_1$ populations G10-3 and G10-4 tested in Lingshui, this QTL showed non-significant effect on TGW and GW (Table 4). In the NIL populations G11-3 and G11-4 tested in Hangzhou, this QTL was detected for TGW and GW with the $R^2$ values ranging from 10.32% to 24.56% (Table 5). Since G11-3 and G11-4 were populations consisting of homozygous lines directly produced from G10-3 and G10-4, the discrepancy is unlikely caused by differences on the genetic background. *Okada et al. (2018)* also mapped a QTL for grain weight and GW in this region, which had a large effect in the late-maturing sub-population but showed non-significant effect in the early-maturing sub-population. They inferred that the difference was caused by the temperature difference from flowering to ripening between the two sub-populations, which was higher for the late-maturing sub-population. Similarly, a stronger effect of *OsMADS51/qHd1* on grain weight was found to be associated with a higher air temperature (*Chen et al., 2018*). The average air temperature from flowering to ripening was 23.5 °C for G10-3 and G10-4, and 27.2 °C for G11-3 and G11-4. This could be the main cause for the inconsistent effect of *qGS10.2* observed in the present study.

Linkage drag is a problem commonly encountered in the introgression of alien genes. *Suh et al. (2011)* reported that lower spikelet fertility was associated with the introduction of the *Bph8* resistance gene from *O. australiensis* into the background of a japonica rice variety. *Yan et al. (2014)* found that transformation of the *Bt* resistance gene into *indica* rice restorer line Minghui 63 may have negative effects on grain weight and grain size, resulting in lower grain yield. One of the parental lines used in the present study, IRBB52, is a rice restorer line carrying *Xa4* and *Xa21* in the genetic background of IR24. In the ZC9 population, three regions other than the targeting region on chromosome 10 were also segregated. QTL for grain size were detected in all these three regions (Table 3), among them *qTGW11* and *qGW11* were located in a region covering the *Xa4* locus (Fig. 2). The IRBB52 allele decreased TGW and GW by 0.14 g and 0.016 mm. These results suggest that linkage drag may have occurred in the introgression of *Xa4* from the donor parent into IR24. Fine mapping of QTL in this region may provide a solution for this problem.

## CONCLUSION

Using NIL populations with sequential segregating regions jointly covering the entire interval for QTL *qGS10* mapped previously, three QTL for grain size tightly-linked on the long arm of chromosome 10 were separated. One of them, *qGS10.1*, had a stable and relatively high effect on grain weight, GL, and GW, providing a good candidate for gene cloning. Another QTL, *qGS10.2*, had a significant effect on all the three traits but the effect was inconsistent across different experiments, providing an example of genotype-by-environmental interaction.

## ABBREVIATIONS

| | |
|---|---|
| **TGW** | 1,000-grain weight |
| **GL** | grain length |
| **GW** | grain width |
| **QTL** | quantitative trait locus |
| **TQ** | teqing |
| **NIL** | near isogenic line |
| **SSR** | simple sequence repeat |
| **GLM** | general linear model. |

### Funding

This work was supported by the National Key R&D Program of China (2016YFD0101104), the National Natural Science Foundation of China (31521064), and a project of the China National Rice Research Institute (2017RG001-2). The funders had no role in study design, data collection and analysis, decision to publish, or preparation of the manuscript.

### Grant Disclosures

The following grant information was disclosed by the authors:
National Key R&D Program of China: 2016YFD0101104.
National Natural Science Foundation of China: 31521064.
China National Rice Research Institute: 2017RG001-2.

### Competing Interests

The authors declare that they have no competing interests.

### Author Contributions

- Yu-Jun Zhu performed the experiments, analyzed the data, contributed reagents/materials/analysis tools, prepared figures and/or tables, authored or reviewed drafts of the paper, approved the final draft.
- Zhi-Chao Sun performed the experiments.
- Xiao-Jun Niu performed the experiments.
- Jie-Zheng Ying performed the experiments.
- Ye-Yang Fan analyzed the data, contributed reagents/materials/analysis tools.
- Tong-Min Mou conceived and designed the experiments.
- Shao-Qing Tang conceived and designed the experiments.
- Jie-Yun Zhuang conceived and designed the experiments, analyzed the data, prepared figures and/or tables, authored or reviewed drafts of the paper, approved the final draft.

### Data Availability

The raw data is available as a Supplemental File.

## Supplemental Information

Supplemental information for this article can be found online at http://dx.doi.org/10.7717/peerj.6966#supplemental-information.

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
