# Peer review of "Dissection of three quantitative trait loci for grain size on the long arm of chromosome 10 in rice (Oryza sativa L.)"

_PeerJ, doi:10.7717/peerj.6966_

## Round 0.1 · original submission · Major Revisions

Both reviewers and I feel that that the manuscript will ultimately be suitable for PeerJ. The major issues are in the presentation and writing. Please address all of the reviewers suggestions. I concur that editing by an native English speaker is mandatory; they are grammatical problems throughout that make the manuscript hard to understand. I strongly urge you to use a professional service

Reviewer 1 ·

Basic reporting

The structure of the manuscript conforms with PeerJ standards and those of the discipline. Raw data are supplied, as required by the journal.

1. However, there are many parts of the manuscript that I found difficult to read and understand, due to grammatical errors, misused words and awkward phrasings. I therefore suggest obtaining assistance from an English-language scientific editorial service. To illustrate what I mean, here are some examples. All of the examples given are from the introduction, but there are similar issues elsewhere in the manuscript.

Line 34: Instead of “All of them are complex traits”, I would say “All of these traits are complex” or “These traits are all complex”

Lines 45-47: Would this be better as “Causal genes have been isolated for 16 quantitative trait loci (QTL) affecting these traits. These loci are distributed across eight of the 12 chromosomes of the rice genome.”

Lines 48-50: I had to pause and reread this sentence to deduce that the ‘opposite directions’ referred to a comparison between effects on GL and effects on GW (at both loci) and not to a comparison between the effects of GW7/GL7 and GS9.

Lines 51-52 and 54-55: The comparisons should be phrased as “larger effects on X and Y than on Z”

Lines 54-55: It should say “have large effects on TGW and GL and no effect on GW”

Lines 58-59 refer to “pathways . . . . , such as ubiquitin-proteosome, phytohormones …….” Some of these may be pathways but others (e.g. phytohormones, transcriptional regulatory factors) are not. This needs to be rephrased.

Line 60: “substantially improved understanding of” or “greatly increased the information available on” would be better. “Extremely” is too extreme. Rather than saying that information has been improved, say that understanding has been improved or that information has been increased.

2. The review of the literature on rice QTL for grain size traits seems good. However, to provide a more complete (and interesting!) rationale for the work, I suggest including some of the context about hybrid rice, which is introduced only in the Discussion.

3. The focus of the Discussion should be on the new results that are presented in this manuscript. Comments on the significance, meaning and novelty of those results can and should be supported by comparisons to other literature, but it always needs to be clear how the points discussed are relevant to the new results. Also, it was not always clear which statements are based on this work and which on the literature. This may improve with language editing.

4. The figures are appropriate but I think that they could be improved. While I don’t think PeerJ specifies line widths or fonts for figures, the use of thicker lines and a sans serif font can make figures easier to read. In Fig. 1, the blue shading of the 5 Mb scale bar is not necessary and is confusing, especially since other shades of blue are used in the legend. I assume that the black rectangles represent centromere positions, but shouldn’t this be specified? Figure 2 is very helpful, but why is one arrow dashed? All of the others are solid. Fig. 4 contains frequency histograms for continuously varying traits. There should be no gaps between the bars, and the the ‘hatching’ of the blue sections of the bars is not necessary. For easier comparison among the 15 charts in Fig. 4, the numbers of lines could be converted to proportions of lines and all charts could be drawn on the same scale (same maximum value on the vertical axes).

Experimental design

This manuscript reports original primary research that is within the scope of the journal. One of its strengths is the rigorous approach taken in materials development.

1. While I understood the research question, I thought it could have been better defined the end of the introduction. The statement about genetic frameworks being lacking puzzled me. With many QTL having been mapped and 16 of them cloned, it seems to me that the genetic framework is pretty good!

2. The statistical methodology could be described in more detail. What was the model that was used in SAS proc GLM?

Validity of the findings

no comment

Reviewer 2 ·

Basic reporting

This paper did QTL analysis using 12 populations in rice, trying to dissect a previously identified QTL region on the long arm of chr10. Three QTL were further refined within this region, with their genetic effects examined in two seperate population groups: F10 pops and F10:11 NIL sets.

This writing is clear and professional, with sufficient background and reference cited. Tables and figures are presented in a clear way. Results were described clearly and reasonable conclusions were drawn.

Experimental design

no comment

Validity of the findings

no comment

Additional comments

L57: grammar mistake: These QTL were found to be...
L65: Please specify what specific trait each group of these QTL involved in
L69: occurred --> been observed
L78: delete "the"
L83: I don't think this is new insight, maybe you want to limit this conclusion to your specific trait and the specific QTL identified here. Also since there is possible very strong GXE between qGS10.1 and qGS10.2, it might worth mentioning.
L102: data not shown), missing a right parenthesis
L105: two others --> the other two
L108: situated --> located
L115: were followed --> followed; same for L121
L125: Which method did you use for QTL mapping? interval mapping or composite interval mapping? please specify.
L176: were used --> used
L184: I don't think you "demonstrated" the observation, maybe using some word less absolute such as "suggested" is more appropriate. same for L191.
L192: I feel expressions such as "a bit", "little" should be avoided in scientific writing.
L206: Since there is such a big discrepancy between your two populations, is it possible to test either of your two hypotheses? I actually doubt phenotypic measurement error could lead to such a big discrepancy.

---

## Round 0.2 · accepted · Accept

Thank you for your careful and thorough revisions. I enjoyed reading about your rigorous approach to dissecting this complex QTL.

# Reviewer 2 ·

Basic reporting

the submitted change fixed all the concerns that I brought up and I agree to accept the current version.

Experimental design

no comment

Validity of the findings

no comment

Additional comments

no comment